# Satellite-derived temperature measures miss key physiologically relevant thermal trends on Palauan reefs

Marilla Lippert[ID]¹*, Maurice Goodman², Brendan Cornwell[ID]³, Katrina Armstrong⁴, Nia S. Walker⁵, Victor Nestor⁶, Yimnang Golbuu⁶, Stephen Palumbi³

**1** Florida State University, Tallahassee, Florida, United States of America, **2** University of Washington, Seattle, Washington, United States of America, **3** Department of Biology, Hopkins Marine Station of Stanford University, Stanford, California, United States of America, **4** University of Maine, Orono, Maine, United States of America, **5** Claremont McKenna College, Claremont, California, United States of America, **6** Palau International Coral Reef Center, Koror, Palau

* mlippert@fsu.edu

## Abstract

Coral reefs are important both economically and culturally to over 1 billion people. However, reefs continue to be threatened by climate change, with some areas now experiencing mass coral bleaching and mortality events due to heat stress on an annual basis. Satellite-derived sea surface temperature data (SSST) are often used as a proxy for in situ temperatures on reefs, and are relied on to identify heat accumulation and assign bleaching risk on reefs worldwide. However, SSST has limitations – readings are only taken at night and on a relatively coarse spatial scale, and multiple studies have exposed discrepancies between SSST and in situ temperatures. In this study, we compare satellite-derived sea surface temperature in Palau to in situ temperature records at 87 reef locations in order to assess how well SSST captures physiologically important thermal trends experienced by corals. We find that while SSST captures average nightly temperatures relatively well, it fails to accurately capture thermal maxima, diurnal range in temperature and heat accumulation measurements like degree heating weeks (DHW) that are relevant in determining coral bleaching risk levels. Though SSST data remain key indicators of temperature stress over global scales, local management of coral reefs, coral restoration, and reef replenishment require more fine scale data in order to accurately understand thermal trends and their implications for coral resilience.

## Introduction

Coral reefs are one of the ecosystems most impacted by climate change, prompting substantial research into characterizing, predicting, and mitigating climate impacts on reefs. For decades, temperature data have been used to assess environmental stress on coral reefs [1]. A wide range of research and conservation

**Data availability statement:** All data and code used to generate analyses and results are permanently archived on Zenodo at https://doi.org/10.5281/zenodo.18303566.

**Funding:** Funding for this project was provided by donors of the super reefs collaborative.

**Competing interests:** Authors have no conflicts of interest to declare.

efforts – including finding and propagating resilient corals [2], locating reefs to serve as climate "refugia" [3], Click or tap here to enter text.tracking heat stress accumulation on reefs [4], and monitoring coral bleaching risk globally [5] – often rely on satellite-derived sea surface temperature (SSST) products [6–8]. Specifically, satellite-derived temperatures have been the base of efforts to produce and distribute warnings of heat stress and bleaching risk [9]. These data are particularly valuable because they are easily accessible, global in extent, and can provide historical temperature measurements going back decades. Given SSST's accessibility, cost, and continuity, it remains a reliable and affordable way to gauge long-term, large-scale trends in ocean temperatures.

However, there are serious limitations to the use of SSST for coral research. First, satellite flyovers provide only one temperature reading per day and during the nighttime, which misses heat conditions during the day and temperature variation (i.e., the difference between thermal maxima and minima experienced in a day by a coral), which is often used as an important indicator of thermal tolerance and resilience in corals [10]. Second, SSST estimates are usually resolved at scales too large (>1km$^2$) to retrieve detailed, accurate information about the heat extremes experienced on smaller reef features such as patch reefs and reef shallows [8,11]. Third, SSST readings are from the topmost layer of the ocean water, which may or may not be representative of temperatures at depths that more accurately represent the thermal stress corals experience [12,13]. As a result of these limitations, caution is often advised when using SSST as a stand-in for in situ bottom temperatures on coral reefs [14,15]. Colin & Johnston (2020) [16], for instance, found that SSST can miss fine-scale and long-term temperature fluctuations across vertical gradients on reefs captured by in situ data. Other studies have found that SSST consistently underestimates temperature across depth gradients [17], and becomes particularly unreliable as latitude increases [18].

Discrepancies between satellite-derived and in situ thermal trends may result in consequences for the efficacy of reef management and the reliability of risk assessments for reefs. SSST is used to calculate a reef's maximum monthly mean (MMM) which is used to inform the coral bleaching threshold (defined as MMM + 1°C) – above which corals experience heat stress and start to bleach. NOAA's Coral Reef Watch (CRW) produces bleaching alerts by observing the degree heating weeks (DHW) of reefs which tracks the frequency and degree to which SSST exceeds the bleaching threshold [19,20]. Degree heating weeks are used to infer heat accumulation, a metric that is highly correlated with coral survival and bleaching [21,22], and is often used to identify areas with potentially more heat tolerant corals. Because maximum mean monthly temperature (MMM) is the foundation of bleaching forecasts, if SSST fails to accurately capture that metric it is likely that the resulting DHW may not be capturing reliable estimates of specific reef heat environments and stress exposure. Over- or under-estimation of heat stress accumulation could have downstream consequences for coral conservation and reef management. Many reef restoration and management groups use heat stress information to guide how they allocate restoration efforts, and where they source more heat-tolerant corals for propagation and

restoration [2]. Accurate recordings of thermal trends on reefs are pivotal to efficient and effective reef restoration efforts, making it increasingly important to understand when and where SSST is a reliable proxy for in situ temperatures.

An alternative to SSST data is datasets derived from in situ temperature loggers. These can provide higher quality and resolution of data because they can take multiple readings each day at the depth of coral colonies, providing more insight into thermal fluctuations experienced by coral colonies [13,23]. This is particularly valuable given the importance of thermal variation in producing more heat-resilient corals [24]. Temperature loggers can also be placed at various reef locations across depth, current, wave or wind conditions, offering highly detailed examination of temperature patterns. However, these advantages come with higher costs in terms of field deployment, testing and collection of the devices.

Because SSST continues to be a popular proxy to characterize the thermal environment that individual corals are experiencing [25–27], it is important to understand the relationship between SSST and in situ datasets. Specifically, it is important to clarify the extent to which physiologically important metrics such as heat stress accumulation and thermal extremes are captured in SSST data [28,29]. In this study, we assess the reliability and accuracy of SSST products for coral reef environmental research by comparing SSST from the National Oceanic and Atmospheric Administration's (NOAA's) Coral Reef Watch (CRW) dataset at a 0.05° daily resolution (ca. 5 km) to an in situ dataset from reefs in three regions across Palau. NOAA's CRW dataset was specifically designed and is currently used for coral reef management purposes such as assigning heat stress levels, flagging areas of high bleaching risk, and tracking long-term thermal trends on reefs [30]. We picked Palau as our study site because of the availability of high resolution, long-term in situ data across the island. By assessing the ability of satellite-derived temperatures to approximate several characteristics of in situ thermal regimes, we aim to inform the appropriate use of these products in the monitoring, protection, and restoration of reef environments.

## Methods

Satellite sea surface temperature (SSST) was obtained from the National Oceanic and Atmospheric Administration's (NOAA) Coral Reef Watch (NOAA Coral Reef Watch). Dates were filtered to match the dates present in the in situ dataset, and location was set to encompass the Northern (a 276 km$^2$ area), Western (a 123 km$^2$ area), and Southern (a 276 km$^2$ area) regions of Palau individually (Fig 1).

In situ data were collected from HOBO temperature loggers (HOBO 64k waterproof logger) zip tied to the reef floor at depths ranging from 1−5 meters on multiple barrier reefs in the Northern, Western, and Southern regions of Palau. Loggers were deployed starting in November 2017 and collected up until January 2021. Temperature data from all three regions were gathered in lagoon reef types with benthic cover dominated by Scleractinian stony corals [32]. The Northern region typically experiences more water movement, potentially leading to different thermal patterns compared to the other two regions, while the Southern region is more sheltered by surrounding smaller islands, and the Western region is moderately sheltered. HOBO loggers drift less than 0.1°C per year according to the product specifications on the manufacturer's website – however, because loggers were replaced on average every 156 days (approximately every 5 months) and no logger was left out longer than 1 year, we did not take drift into account. In total, we collected data from 87 HOBOs: 35 in the Northern region, 26 in the Western region, and 26 in the Southern region (Fig 1). HOBOs were set to record temperatures at 10-minute intervals. To ensure comparability between SSST data – which is measured at night – and in situ measurements, HOBO data were filtered to contain only nighttime temperatures (between 10:00PM and 2:00AM) for most analyses. Nightly and monthly means, nightly maximums, and nightly minimums for each region (Northern, Western, and Southern) and the overall region (denoted "all") were used to compare to satellite-derived temperatures. HOBO loggers also have an accuracy of +/- 0.53°C according to the product specifications on the manufacturer's website, however we greatly reduce the influence of this intrinsic random noise by grouping and averaging our HOBO logger data across regions and having multiple loggers per satellite grid cell. Because HOBOs were launched on land, all data prior to one day post HOBO deployment on the reef were discarded so as to not capture air temperature recordings. Gaps in HOBO data reflect field seasons where temperature loggers were retrieved and/or redeployed.

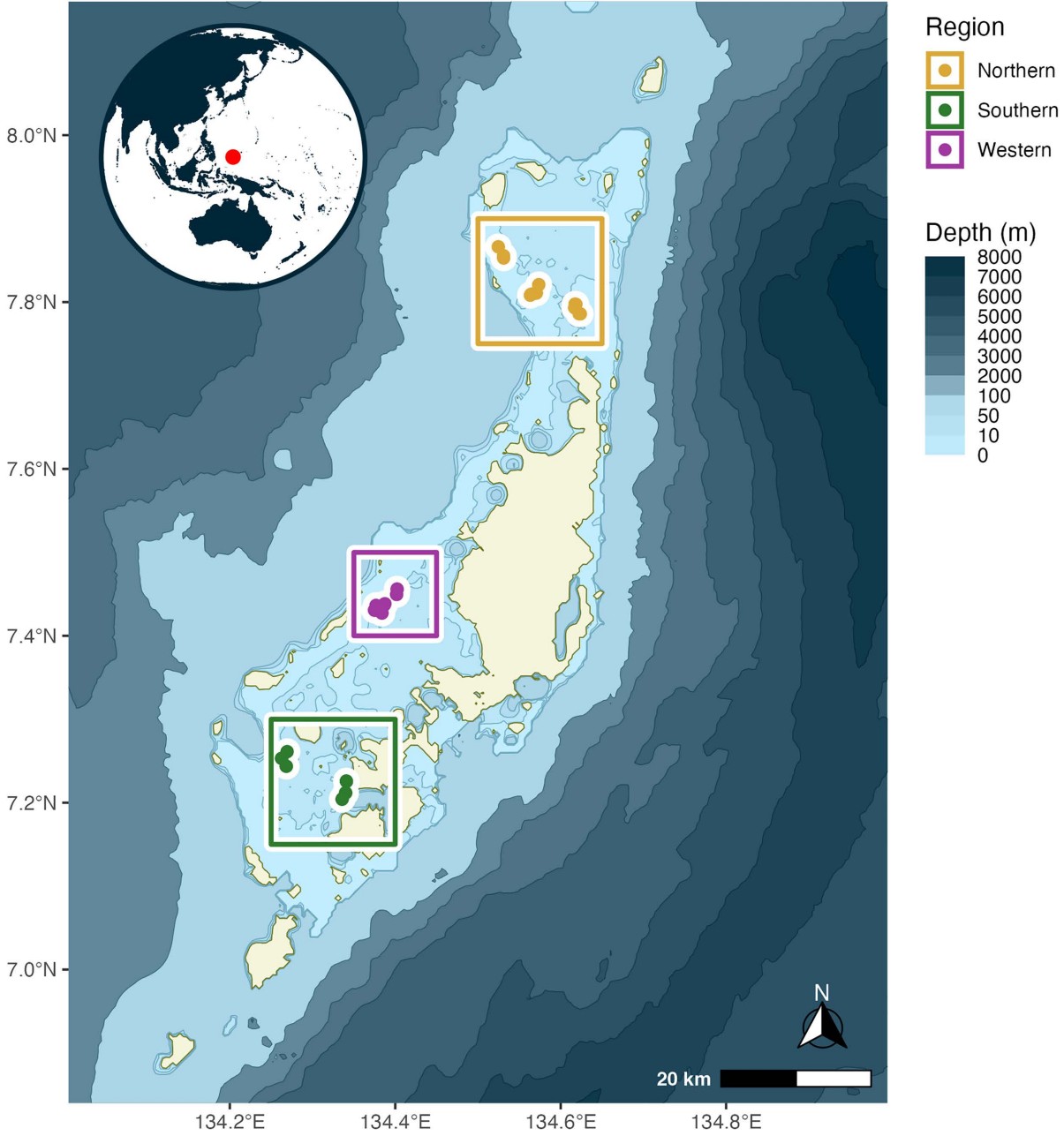

**Fig 1. Palau site map.** Map of locations where in situ loggers were located (circles) and bounding boxes from which SSST was extracted from (squares) for the study. Inset map shows the location of Palau in the greater Pacific region. Bathymetry and coastline layers extracted from NOAA's ETOPO 2022 15 Arc-second global relief model [31]. Inset made with Natural Earth.

Maximum monthly means (MMM) – defined by NOAA as the warmest month mean of the previous 12 months – for in situ and SSST datasets were calculated according to NOAA's CRW methods [33]. Bleaching thresholds – defined by NOAA as the MMM plus 1°C – were calculated for each method (in situ and SSST) [34]. Heat accumulation measured in degree heating weeks (DHW) was calculated using the 12-week sliding window method reflected in NOAA's CRW DHW

assignments. For each 12-week window, the DHW is assessed by summing the magnitudes of instances where temperatures exceed the bleaching threshold. The sliding window then progresses one day at a time and recalculates the DHW in each instance. In order to see if SSST data could be used to infer the thermal variability of a reef, we assessed whether there was any correlation between the nightly average temperature recorded by satellites and daily thermal range (maximum – minimum °C) detected by in situ loggers.

R version 4.4.1 was used to conduct data management, visualization, and analysis [35]. For each metric described above, we compared in situ-derived values to SSST-based estimates (nightly and monthly means, nightly maximums, and nightly minimums) graphically and using separate linear regression models (with in situ temperatures as the response variable) and Pearson correlations for the full dataset and for each region (Northern, Southern, and Western) separately (Fig 2). Because we expected fluctuations in daily and monthly SSST to be associated with changes in in situ temperature *a-priori*, we constructed tests of the SSST slope coefficient with a null of $\beta_0 = 1$ (instead of the default null of $\beta_0 = 0$) to test specifically whether the estimated relationship between SSST and in situ temperature was consistent with the null expectation that a 1°C change in SSST is associated with a 1°C change in in situ temperature in the same direction. Therefore, if neither the intercept nor the slope differ significantly from 0 and 1 respectively, then for every 1°C increase in SSST, in situ temperatures also increase by 1°C and there is no consistent bias (intercept = 0). If the intercept is significantly different than 0, then SSST-derived metrics are on average higher (intercept < 0) or lower (intercept > 0) than in situ temperatures. Likewise, if the slope is significantly different than 1, then changes in SSST-derived metrics dampen (slope > 1) or exaggerate (slope < 1) changes in in situ-derived metrics. Models were consistently characterized by high levels of residual temporal autocorrelation, so we corrected coefficient standard errors (and thus t-statistics and p-values) by applying the Newey-West heteroskedasticity and autocorrelation consistent estimator to the coefficient variance-covariance matrix for each of the regional models [36,37]. For models using data from all regions, we used the Driscoll and Kraay panel corrected covariance matrix estimator, a generalization of the Newey-West estimator for data repeated across grouping levels/ regions [38]. DHW regressions were computed for the full region without examining sub-regional differences due to gaps in the regional datasets. Where tests were computed at both regional and sub-regional scales, we also corrected for multiple comparisons by applying the Bonferroni correction to each family of four tests (the full model and models for each of three sub-regions) corresponding to a given temperature metric.

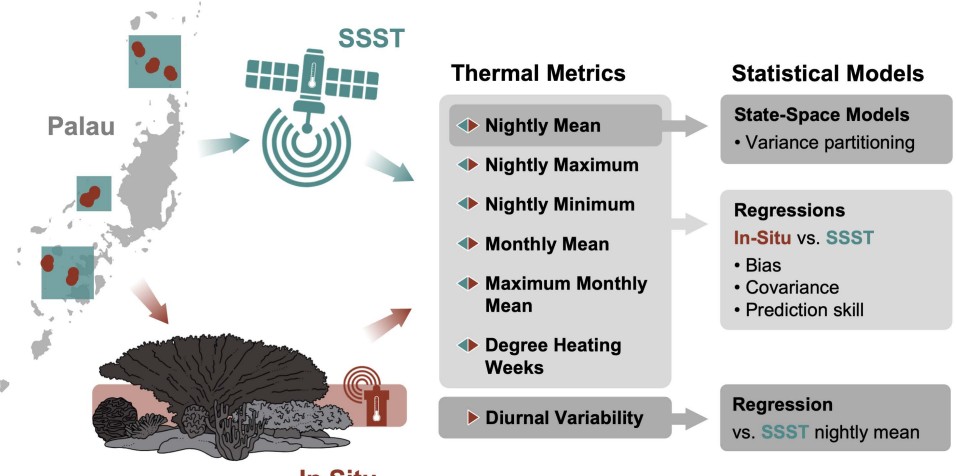

**Fig 2. Methods flowchart.** Diagram showing where and how datasets were generated, what thermal metrics were obtained, and what statistics were run. Inset made with Natural Earth.

We also used the above detailed regression models to evaluate the skill with which (given regional time series of both in situ and SSST data) SSST could be used to predict in situ data. To do so, we used standard model performance metrics ($R^2$ and Root Mean Square Error [RMSE]) to evaluate in-sample skill, and a walk-forward validation approach with an expanding window to evaluate out-of-sample predictive skill. In the walk-forward approach, we trained each region- and metric-specific model on all available data up to a forecast origin at time $t$, used the model to predict all daily in situ temperatures from SSST data up to time $t+30$, and calculated the Mean Absolute Error (MAE) for this 30-day horizon. We then shifted the window forward by 30 days, repeated the evaluation, and averaged the resulting set of MAE values. To avoid predicting over data gaps, we only predicted for horizons within the same time block as the forecast origin, and we used April 2019 as the first forecast origin.

To examine the extent to which trends in each of the six (two data origins in each of three regions) nightly mean temperature time series reflected variability shared among all time series, shared between data origins or regions, or unique to each time series, we implemented two multivariate state-space models using the "MARSS" R package [39]. In the first, we modeled each time series as arising from one of three regional latent states, where the model estimated variance both at the regional level and shared among regional states, corresponding to the hypothesis that temperature trends are distinct in each region and that regional in situ and SSST data measure these states with error. In the second, we modeled each time series as arising from either an in situ or SSST latent state (partitioning variance in the same way as the first model) corresponding to the hypothesis that in situ and SSST data are instead measuring distinct latent trends (i.e., sub-surface and surface) that are more similar among regions than they are between different data origins within the same region. Each model included as a response all nightly average temperatures for each time series, standardized to have zero mean and unit variance. We estimated latent states as random walk processes, with separate observation error terms and state loadings for each time series.

## Results/Discussion

### Average temperatures

We compared SSST with in situ data from 87 temperature loggers from late 2017 to early 2021 across a wide range of reefs in Palau. For comparability among data sources, most statistics and derived quantities use nighttime in situ temperatures, although we note average daily (24-hour) in situ temperature is highly correlated with SSST reads ($r=0.92$, $r \in 0.93-0.94$, S1 Fig, S1 Table).

Average nightly in situ and satellite-derived temperatures were highly correlated across all regions ($r=0.90$, $r \in 0.89-0.93$). Despite these high correlations, satellite-derived temperatures were biased low, reading on average 0.58°C lower than in situ temperatures (S1 Table, Fig 3). In addition, SSST data underestimate nighttime in situ temperatures by 1°C or more on multiple occasions in each location (Fig 3). We also note that there are instances in all locations, seasons, and years when SSST data over-estimate in situ conditions, though these instances are rare (less than 2% of days in the overall region, Fig 3).

The deviation of SSST from in situ data is inconsistent and differs depending on what scale and where the comparison is made. For example, in the Northern region, SSST records on average 0.39°C cooler than in situ loggers – the smallest difference we see – while in the Southern region SSST records 0.58°C cooler on average. In the Western region, where reef temperatures are generally higher, SSST reads 0.71°C cooler than in situ loggers (S1 Table). While these differences are notable, neither the slope nor intercept were significant for any individual region, indicating that on a smaller, regional scale, SSST can reliably be used as a proxy for the average in situ nighttime temperature. However, in the overall region ("all"), the slope was significantly greater than one, indicating that on a larger scale (greater than 10 km) in situ mean nightly temperatures are more variable than SSST reads.

These results suggest that SSST on its own is an unreliable predictor of average in situ nighttime temperatures, and a period of known in situ nighttime temperature is necessary to inform the relationship between SSST and in situ data

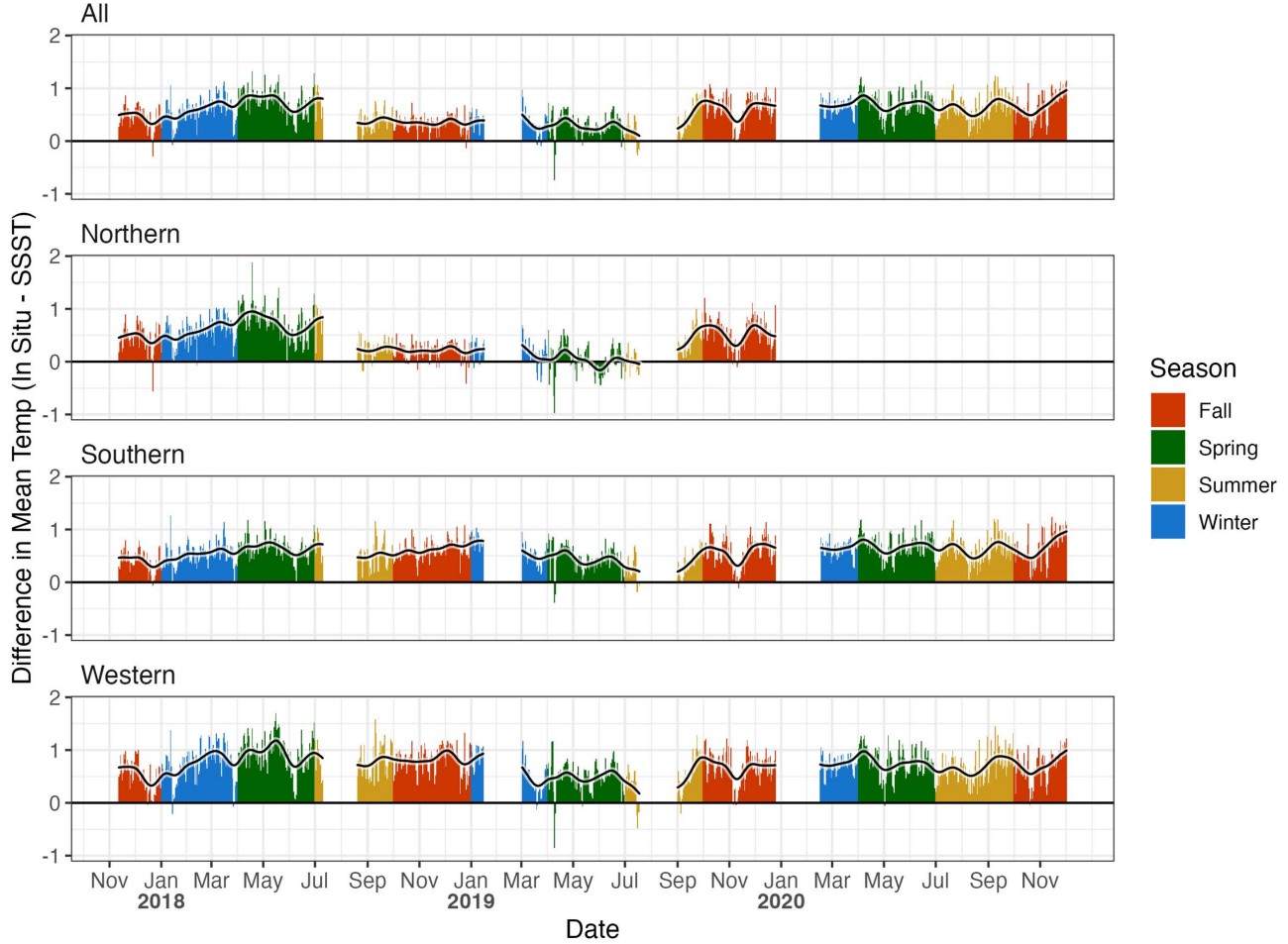

**Fig 3. In situ and SSST disparities.** Disparity between in situ and SSST nightly average temperatures across all regions, seasons and years. Positive values indicate in situ readings were higher than SSST readings, and negative values indicate the times when in situ values were lower than SSST readings. The superimposed black line is a Gaussian kernel smoother with a 1-month bandwidth. The majority of differences are positive across all regions, seasons, and years, indicating that SSST regularly reads lower than in situ temperatures.

sources, and to correct SSST bias (Figs 4 and 5). SSST data are often used to make statements about broader regions (i.e., at the island or large reef system scales) [3,40,41], but local calibration using nighttime logger data may be necessary when using SSST as a proxy for in situ temperatures on the reef floor. Predicting out-of-sample mean in situ monthly temperature using SST substantially reduces predictions errors (MAE $\in$ 0.1–0.2°C) compared to using the mean of monthly means (MAE $\in$ 0.5–0.55°C), suggesting local calibration may resolve discrepancies on coarse frequencies. Improvements are more marginal for nightly mean in situ temperature, with MAE $\in$ 0.25–0.36°C for out-of-sample prediction using SSST compared to MAE $\in$ 0.55–0.59°C using the mean of nightly means (Fig 5). While seemingly small, these differences may lead to compounding errors since these temperatures are used to estimate bleaching thresholds, resulting heat stress, and bleaching risk. Because SSST is used to determine bleaching risk, these deviations may lead to meaningful differences in the heat stress levels assigned to reefs, particularly when nightly average temperatures are more extreme. This could in turn lead to differing recommendations for management and mitigation during heat stress events.

State-space model fits to nightly average temperatures confirm these findings. Fit to the full dataset was substantially better for a model where variation in underlying (latent) trends differed by data origin (AIC = 407.0) than a model

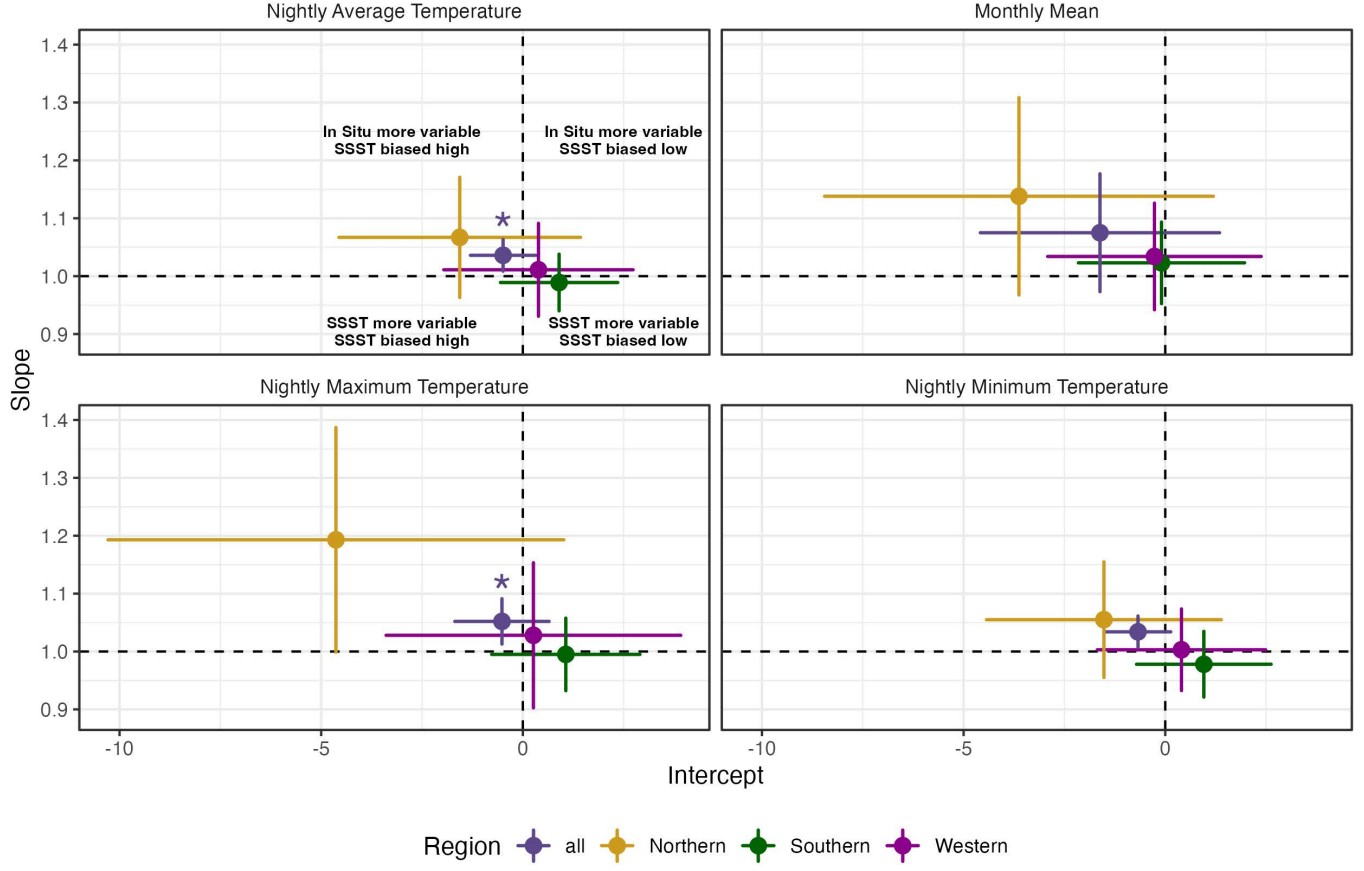

**Fig 4. In situ and SSST estimates.** Regression coefficients (slope and intercepts) for the relationship between in situ and SSST-derived estimates (with in situ temperatures as the response variable) of nightly mean, nightly maximum, nightly minimum, and monthly mean nighttime temperatures. Points are colored by region. Significant estimates (either the slope or intercept, $p < 0.05$) are indicated by an asterisk in the same plane as the estimate. Lines represent the 95% confidence intervals of the respective estimates. Points that lie above the dashed line at slope = 1 have more variation in in situ temperature, while points below have more variation in SSST data. Points that lie to the left of the dashed line at intercept = 0 show where SSST is biased high, and points to the right show where SSST is biased low. Cases where the confidence interval lines cross the dashed line imply no detectable difference in variability or bias between methods.

where each data origin measured differing latent regional trends (AIC = 1245.6), suggesting that in situ and SSST data are measuring different processes and that temporal variability in in situ data more closely mirrors in situ data from other regions than collocal SSST data. Notably, estimates from the regional model (S2 Fig) substantially dampen the magnitude of shared variability among in situ datasets, while the data origin model closely approximates trends in most regions (S3 Fig). In the data origin model, data origin explains 45% of variance in temporal trends across all time series (compared to 1% explained by region in the regional model), with 14% attributable to variation shared between in situ and SSST data. Together, these results suggest that average nightly temperatures measured in situ capture dynamics that are shared among regions and not measured by SSST.

### Inter-regional comparisons

Regional comparisons with in situ data (Fig 6a) and SSST data (Fig 6b) show general agreement between methods in terms of directional differences between regions, but more extreme and frequent differences between regions are captured when using in situ data. Both methods show the Western region is on average and more often warmer than the Northern

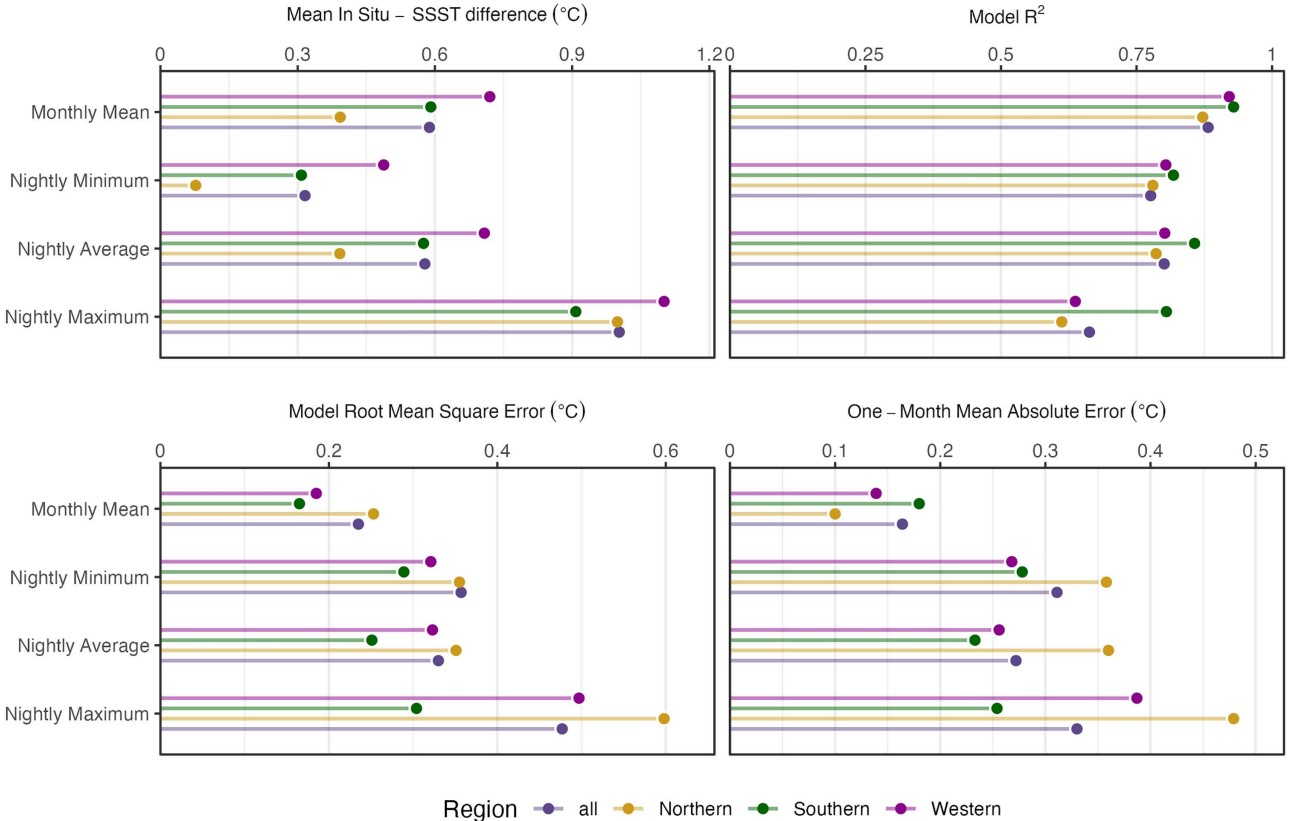

**Fig 5. Performance metrics for prediction of in situ data from SSST.** Panels show mean difference between in situ and SSST derived temperature estimates, $R^2$ and Root Mean Square Error (RMSE) from in situ vs. SSST regressions, and Mean Absolute Error (MAE) of one-month walk-forward predictions of in situ temperature given known SSST. Performance metrics are shown for nightly mean, nightly maximum, nightly minimum, and monthly mean nighttime temperatures (rows) in all regions (colors).

and Southern regions, but the average difference between the regions is larger for in situ data (0.36°C) than the SSST data (0.06°C). In addition, the Western region is warmer than the Northern region 88% of the time according to in situ data, and only 69% of the time according to SSST data. Similar trends are seen in the Southern and Northern regions – both methods agree the Southern is on average and more frequently warmer, but the average difference is 0.2°C higher in in situ data. When comparing the Southern and Western regions, there is complete disagreement between methods. In situ data shows the Western region is warmer than the Southern region by 0.1°C on average and 67% of the time, while SSST data shows the Southern region is warmer than the Western region by 0.05°C on average and 72% of the time. While state-space model estimates suggest in situ and SSST data capture different trends which are shared among regions, they also suggest that in situ data display greater variation among regions than do SSST data – on average, 51% of variability in individual in situ time series is unexplained by shared variation with other in situ or SSST time series, compared to 30% for SSST time series (S4, S5 Figs). Overall, we found that general temperature differences between regions are more pronounced in the in situ data versus the satellite-derived data (Fig 6). These regional differences are particularly relevant for coral reef restoration groups that may maintain in situ coral nurseries. Site selection criteria for coral nurseries are extensive and often include thermal criteria such as maximum temperatures [42]. SSST is a common data source used to determine the thermal suitability of sites due to its accessibility and low cost. If more fine-scale, regional thermal trends are not accurately captured, this could lead to inefficient use of resources, and potentially decreased survival of coral outplants in the nursery.

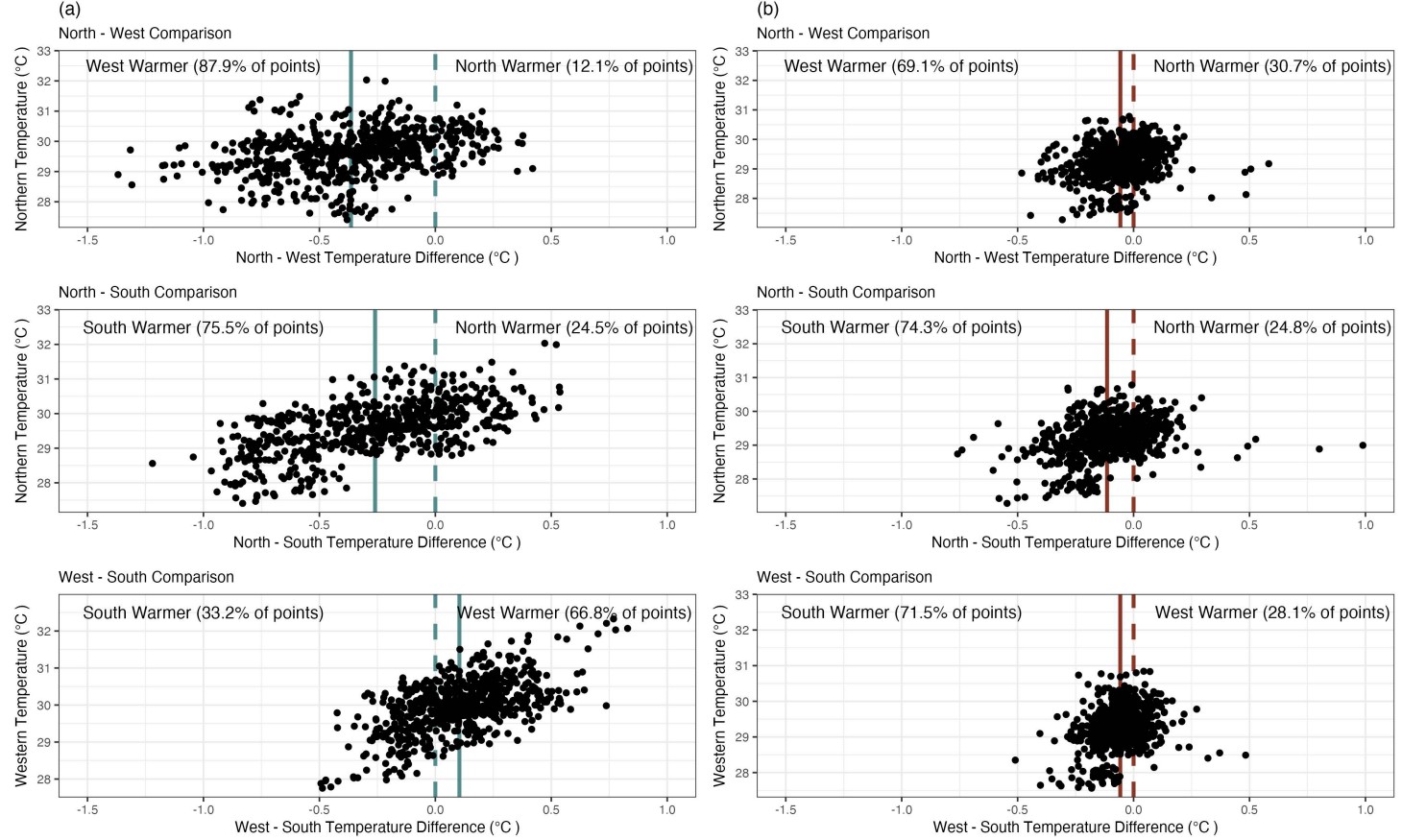

**Fig 6. In situ and SSST regional comparisons.** Differences in regional temperatures recorded between in situ (A) and satellite (B) methods. The dashed lines mark zero, where there is no difference in temperature between compared regions, and the solid line represents the mean difference in temperature between compared sites. Each point represents the difference in temperature on a specific day. Percents next to "x warmer" labels indicate the percent of points (days) where region "x" was warmer.

## Thermal extremes

Previous studies have shown that coral responses to heat stress can vary in response to past heating and bleaching events, and that reefs that experience higher temperatures more frequently have a higher rate of coral bleaching and mortality [43,44]. In addition, it has been shown that regions and coral species can have differing thermal thresholds, where risk of mortality greatly increases if the coral is exposed to temperatures that exceed that threshold [45,46]. Therefore, to accurately interpret risk level for reefs it is necessary to have reliable estimates of thermal maxima for a reef.

Nightly maximum temperatures were an average of 1°C higher for in situ data than SSST data (S1 Table). Overall correlation between in situ and SSST maximum nightly temperature was slightly less correlated than average nightly temperatures but still relatively high (r = 0.81), while the Northern region showing the lowest correlation (r = 0.78), followed by the Western and then Southern correlations (r = 0.80 and r = 0.90, respectively) (Fig 4, S1 Table). However, the SSST slope for the overall region was significantly greater than one (Fig 4), indicating maximum in situ temperatures were more variable than SSST reads, and that maximum in situ temperatures cannot be inferred from satellite temperatures without prior and site-specific knowledge of the in situ-SSST relationship. Model fit is generally worse ($R^2 \in 0.61$–$0.8$) and out-of-sample predictive error is generally higher (MAE $\in 0.25$–$0.48$°C) for nightly maximum temperatures than for other

temperature metrics, suggesting limited potential to resolve discrepancies between SSST and in situ derived temperature maximums using local calibration as compared to mean temperature (Fig 5).

We also compared the nightly minimum temperatures between methods and found similarly high correlations overall (r = 0.88) and regionally (r ranging from 0.88–0.90). On average, SSST recorded minimum temperatures 0.32°C cooler than in situ temperatures overall (S1 Table). In all comparisons, neither the slope nor the intercept was significant, indicating that SSST temperature could accurately be used to predict minimum in situ temperatures at a larger (overall) and smaller (regionally) spatial scale (Fig 4). While such differences in measurements of thermal extremes between methods may not be surprising, they are consequential because thermal extremes, especially maximums, are important for determining coral bleaching and mortality risk [11,47,48]. Inability to capture accurate thermal maxima on reefs will lead to inaccurate estimates of heat stress accumulation in corals on those reefs. As with other discrepancies discussed between data sources, this misalignment may have significant negative impacts on coral reef restoration practitioners and managers who rely on accurate estimates of heat stress and thermal trends to inform their restoration and mitigation strategies. Such discrepancies should not be ignored, especially when data during abnormally warm periods are used to assign bleaching risk and alert levels to reefs [49].

## Thermal variability

Thermal variability on reefs may enhance coral resilience to heat stress by promoting plasticity, thus reducing the risk of coral bleaching (Safaie et al., 2018; Thomas et al., 2018). Daily temperature variation can be a strong indicator of coral stress (Oliver & Palumbi, 2011), and can signal changes in coral transcription on an hourly basis (Ruiz-Jones & Palumbi, 2017). Environments with high thermal variability tend to house corals with higher growth rates and increased ability to handle heat stress [50], while reefs that experience more frequent thermal fluctuations have been shown to house corals with higher heat resistance [11,51]. In addition, restoration practitioners will often target thermally resistant corals to use as propogation sources for outplanting onto the reef during restoration efforts. Because thermal variability is such an important metric, it is important that we understand if SSST, despite being derived from a once-daily dataset, can provide any insight into the thermal fluctuation's corals experience. We therefore regressed daily (24-hour) measures of temperature variance in in situ data against average nighttime SSST to assess whether satellite-derived temperatures can be used to infer in situ thermal variability.

Our analyses show that in situ records of diurnal temperature variability are poorly correlated with SSST data (r = −0.01 overall, Fig 7, S1 Table). Regional correlations between methods were also poor, ranging from 0.06 to 0.1. All scales (overall and regionally) had significant slopes, and the overall region also had a significant intercept. These results indicate that there is no reliable relationship between SSST average nightly temperature and diurnal variability, and no way to infer diurnal variability from SSST. While this result may be expected, temperature variability is a key feature of coral thermal adaptation [24] and if SSST is to be used to predict coral sensitivity and resilience to thermal stress, it should be acknowledged that SSST does not and cannot capture a key component of what indicates heat tolerance.

## Heat accumulation and bleaching risk

Maximum mean monthly temperature (MMM) is the foundation of bleaching forecasts. Because month means are on average 0.6°C different (S1 Table), it is likely that resulting SSST derived DHW may not be able to detect accurate estimates of heat stress accumulation on these reefs. To assess agreement in DHW measurements between data sources, we used NOAA's CRW methodology to calculate bleaching thresholds and DHW for both in situ and SSST data in every region (All, Northern, Western, and Southern). Monthly mean temperatures were highly correlated between in situ data and SSST data (r = 0.94, S1 Table), with no significant variable in any comparisons (Fig 4), indicating that at small and large scales, SSST can reliably be used to infer in situ month mean temperatures, and by extension the maximum monthly mean. However, in order to not inflate differences between methods, the MMM for in situ data was adjusted (MMM in

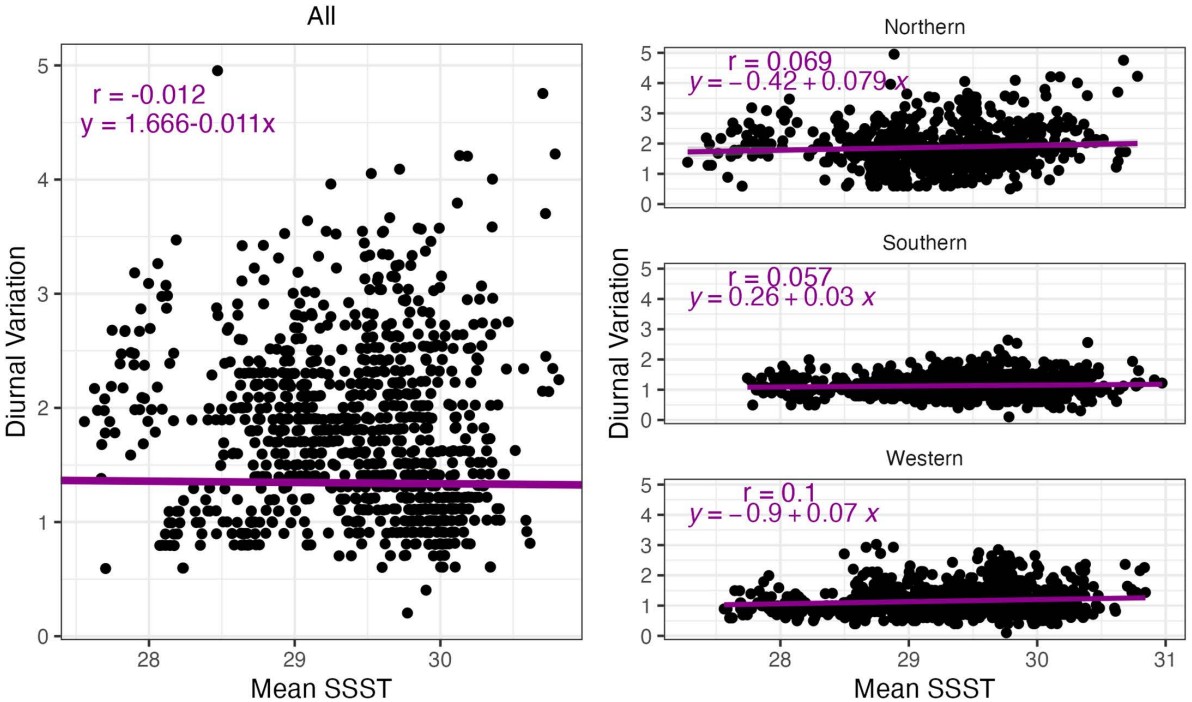

**Fig 7. Diurnal variation versus mean SSST.** Comparison of mean nightly SSST temperatures and diurnal variation detected by in situ loggers, broken into All, Northern, Southern, and Western regions. Each point represents a day, and the colored lines show the trendlines for the relationships. Pearson correlation coefficients (r) and best-fit regression line coefficients are denoted in each panel.

situ = SSST MMM + average month mean offset). To calculate the heat accumulation measured by each method, we took the nightly average temperatures and summed the degrees over 1°C over their respective MMM's (for SSST = 29.23°C, in situ = 29.81°C) across a retrospective 12 week window to get DHW. For this comparison, we limited our study window to 2020 because it has the longest stretch of continuous data.

Our results show that in situ and SSST capture significantly different signals of heat stress accumulation (Fig 8). While both methods agree when there is no heat stress accumulation (DHW = 0, intercept p-value = > 0.05), in situ data showed significantly more heat stress accumulation and higher DHW than SSST data (slope p-value = < 0.005) (Fig 8c). In situ temperatures log more extreme and more frequent incidences where reefs exceed the bleaching threshold compared to those logged by SSST (Fig 8a). There was general agreement between methods on when heat stress accumulated and diminished, but the rate of heat stress accumulation for in situ temperatures is significantly faster than those detected by SSST (Fig 8b). The disagreement in the extent of heat accumulation detected by our data sources is particularly important as heat stress accumulation estimates are widely used in coral bleaching risk and mortality predictions [22].

Discrepancies between in situ and SSST data may lead to underestimating bleaching risk of reefs, or misidentifying reefs that may house more resilient corals, potentially leading to differing conclusions depending on the data source used. In Palau, for instance, two studies were conducted examining where heat resistant corals could be found: one study sourced its temperature data from in situ loggers and found higher concentration of heat resistant coral colonies in warmer micro-habitats [11], while the other study sourced its temperature data from SSST and found more heat resistant corals in thermal refugia, or less warm waters [8]. These two studies highlight the importance of scale and intentionality when selecting whether to use SSST data or in situ data.

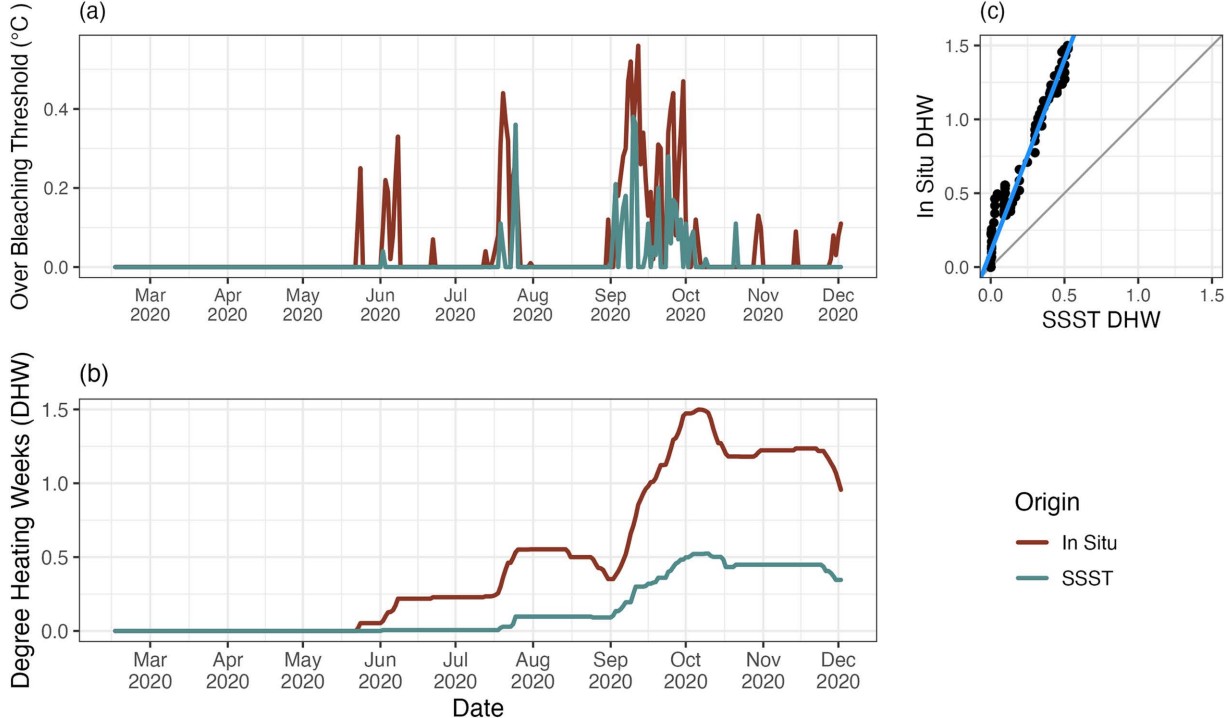

**Fig 8. Heat accumulation comparison.** Comparisons of heat accumulation metrics (degrees over bleaching threshold and degree heating weeks (DHW)) in 2020 for each data source with in situ in red and SSST in blue. Comparisons are for the whole region (all). **(A)** Extent that each method (in situ and SSST) exceeded their respective bleaching thresholds in 2020. **(B)** DHW logged by each method on a 12-week sliding window scale. **(C)** Direct comparison of in situ DHW and SSST DHW, where the grey line is the 1:1 line and the brown line is the fitted trendline. SSST data logged significantly fewer DHW's (or less heat accumulation) than in situ data.

However, while some studies have shown that corals from sites with more thermal variation are more heat resistant in lab-based heat stress experiments, this does not always appear to be the case in nature. A study conducted in Australia surveying coral cover on thermally variable and stable reefs before, during, and after a bleaching event found that coral cover decreased more in thermally variable sites than in thermally stable sites [52]. This same study also revealed that temperature trends and the relationship between thermal variability and loss of coral cover post-bleaching differed by reef types (i.e., reef slope vs flat vs crest vs lagoon), although thermally stable reefs consistently retained higher coral cover post-disturbance. Higher rates of coral cover loss results in larger shifts in the benthic composition of the reef, often to more algal-dominated benthic communities [53]. Understanding when thermal extremes may result in broader shifts in benthic communities can help local restoration and conservation organizations better prepare for the potential results of heating events and allocate their resources to support more informed, targeted mitigation strategies.

Many reefs have complicated topographies with reef crests, lagoons, flats, and slopes often being relatively close together spatially. Because the majority of the reefs we deployed in situ temperature loggers on were reef flats, the relationships we found between in situ and SSST datasets may therefore only be applicable to this specific reef types, further complicating the relationship and reliability of SSST as a proxy for in situ temperatures. SSST may not always be able to provide the resolution of data needed to accurately capture important fine-scale thermal patterns for predicting heat resistance in corals. These discrepancies may have large implications for restoration groups that often utilize such studies to determine where heat resistant colonies are likely to be located to determine where they should focus protection or source

restoration projects from. Future work could aim to clarify the relationship between in situ and satellite-derived temperatures in different reef types, and the extent to which thermal variability results in in situ bleaching resistance.

## Conclusion

Overall, we find that SSST can accurately capture some important characteristics of thermal regimes such as mean and extreme temperatures, but that the accuracy of SSST may depend on the scale to which it is applied. Additionally, we found that local calibration using a period of in situ data to correct SSST bias may allow for predictions of in situ temperature with generally low (< 0.4°C) error for mean nightly and monthly temperatures. However, we found that SSST variation generally dampens variability (and thus estimates of thermal extremes) seen in in situ datasets in all regions, with greater inter-regional covariance among in situ temperatures than between SSST and in situ temperatures within regions. These discrepancies, while small, lead to persistent biases in SSST-derived approximations of physiologically important derived thermal indices.

Our study revealed that SSST data cannot be relied on to capture some key thermal metrics relevant to coral heat resilience and bleaching threat. Significant differences in maximum temperatures detected at a broad scale resulted in the inability of SSST data to accurately determine the amount of heat stress accumulation on reefs. SSST severely underestimates DHW, a key indicator of bleaching risk used to inform reef management. While the affordability and ease of access of SSST makes it an invaluable resource for a global-scale comparative framework, we find that in situ loggers are a more accurate way to measure exposure of individual reefs or corals to local heat stresses, and may be more appropriate for informing local coral reef management and protection strategies in terms of identifying areas at risk and areas of thermal refuge. Our study highlights cases where the relationship between SSST and in situ temperatures are consistent and predictable, but require in situ data to inform the magnitude and direction of correction needed to adjust SSST data. While it is oftentimes impracticable to deploy and maintain in situ loggers on reefs full-time, it is possible that in situ loggers may only need to be deployed for a few months to arrive at bias-corrected regional scale satellite-derived temperature metrics, overcoming dataset limitations.

Further studies are needed to understand 1) how much prior data is needed to sufficiently inform the relationship between in situ and satellite-derived temperature, 2) how long that relationship remains accurate or whether or not it is seasonally dependent, and 3) how the relationship between the two methods will change as global climate trends continue to shift. Climate restoration and mitigation efforts rely on understanding and tracking the physical environment, making it increasingly important to ensure our data can produce accurate information. As global temperature increases and oceanic heat waves become more common and intense, it is important we continue to assess the reliability and accuracy of our environmental data sources to ensure that our marine restoration and mitigation efforts are well informed and have higher chances of success.

## Supporting information

**S1 Table. Pearson correlations between in situ and SSST for all, northern, western, and southern regions as well as the general least squares model results with in situ as the response variable, a slope assumption of 1, and an intercept assumption of 0.** Bonferroni correction applied for a family of 4 tests. Newey-West/Beck & Katz correction applied to regional/full SEs.
(PDF)

**S1 Fig. Average nightly (between 10 pm and 2 am, blue) and average daily (24-hour, orange) in situ data compared to nightly mean SSST.** Colored lines are trendlines. Pearson correlation values are denoted in the color of the comparison.
(TIF)

**S2 Fig. Time series of standardized observed temperatures (blue) for each region (rows) and data origin (columns), with fitted estimates (black) and 95% confidence bands (grey) from a state-space model which links variation in each time series to one of three latent regional states.**
(TIF)

**S3 Fig. Time series of standardized observed temperatures (blue) for each region (rows) and data origin (columns), with fitted estimates (black) and 95% confidence bands (grey) from a state-space model which links variation in each time series to one of two latent states by data origin.**
(TIF)

**S4 Fig. Time series of estimated latent states and 95% confidence bands from a state space model where states differ by data origin.**
(TIF)

**S5 Fig. Proportion of variance explained by different sources for each region (rows) and data origin (panels).**
"Origin" refers to variation shared with other regional time series of the same data origin, "Inter-origin" refers to variation which is shared between in situ and SSST data, and "Obs. error" refers to variation specific to individual region- and data origin-specific time series that is not attributable to shared variation with other time series.
(TIF)

## Acknowledgments

The authors would like to thank the staff and boat operators at the Palau International Coral Reef Center for providing facilities and resources to collect this data. We would also like to extend our gratitude to the state governments of Aimeliik, Kayangel, Koror, and Ngarchelong for their support in this project and the deployment of our temperature sensors.

## Author contributions

**Conceptualization:** Marilla Lippert, Stephen Palumbi.

**Data curation:** Katrina Armstrong, Nia S. Walker, Victor Nestor, Yimnang Golbuu, Stephen Palumbi.

**Formal analysis:** Marilla Lippert, Maurice Goodman.

**Supervision:** Stephen Palumbi.

**Visualization:** Marilla Lippert.

**Writing – original draft:** Marilla Lippert.

**Writing – review & editing:** Maurice Goodman, Brendan Cornwell, Katrina Armstrong, Nia S. Walker, Stephen Palumbi.

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
