## [Decision Letter · Decision Letter 0]

17 Nov 2025

Dear Dr. Lippert,

Thank you for submitting your manuscript to PLOS ONE. After careful consideration, we feel that it has merit but does not fully meet PLOS ONE’s publication criteria as it currently stands. Therefore, we invite you to submit a revised version of the manuscript that addresses the points raised during the review process.

We look forward to receiving your revised manuscript.

Kind regards,

Parviz Tavakoli-Kolour

Academic Editor

PLOS ONE

Journal Requirements:

“Funding for this project was provided by donors of the super reefs collaborative.”

4. Please include captions for your Supporting Information files at the end of your manuscript, and update any in-text citations to match accordingly. Please see our Supporting Information guidelines for more information: http://journals.plos.org/plosone/s/supporting-information .

Reviewers' comments:

Reviewer's Responses to Questions

**Comments to the Author**

1. Is the manuscript technically sound, and do the data support the conclusions?

Reviewer #1: Yes

Reviewer #2: Yes

2. Has the statistical analysis been performed appropriately and rigorously?

Reviewer #1: Yes

Reviewer #2: No

3. Have the authors made all data underlying the findings in their manuscript fully available?

Reviewer #1: Yes

Reviewer #2: Yes

4. Is the manuscript presented in an intelligible fashion and written in standard English?

Reviewer #1: Yes

Reviewer #2: Yes

Reviewer #1: Overview

Overall, this manuscript presents an interesting and valuable story about comparing in situ and satellite-derived thermal information and their implications for coral reef management. The major claim of this paper relates to the limitations of satellite-derived thermal measurements to provide information at management-relevant scales. The manuscript is well written and flows in a way that is easy to follow and understand.

However, there are a few things that need some attention. First, the title suggests that reef physiology (e.g., accretion, benthic composition, primary production, etc.) is examined in the manuscript as a response variable, but this is not really the case – it’s more about thermal stress and temperature as an environmental variable at different scales. Second, some terms that are discussed in the results/discussion section need to be presented earlier both in the introduction and methodology sections. Some rewording is also necessary across the different sections to reduce repetition and increase clarity.

I recommend the manuscript to be considered for publication once the revisions have been incorporated.

Introduction

This section does a good job at highlighting the relevance of the research study and putting it into context. Some rewording and adding references to support some claims would be beneficial.

Methods

This section is very detailed and provides enough information regarding data origin and management. It also provides good information about the statistical analyses. However, some rewording is needed as robust analytical methods are presented and sometimes become hard to follow.

More detail about the composition in terms of reef characteristics (reef types, geomorphology, overall benthic composition/communities, etc.) might help to understand the differences between the regions (Northern, Southern, Western) and put into perspective the importance of finer-scale thermal data.

Results/Discussion

Overall, this section does a good job at presenting and summarizing the main findings of the study. However, if results and discussion are joint into one single section, the authors need to ensure the results are discussed more thoroughly and put into context as they are presented.

Average temperatures

This section does a good job at summarizing the comparison between average in situ and SSST measurements. However, it needs to put more into context what these differences represent: 0.39/0.58/0.71/1°C difference seems small, but is it? What does it mean for coral reefs and managers?

Inter-regional Comparisons

This section does a good job at highlighting the differences across regions, but, as the section before, it needs to put discuss what this means for coral reefs and for management.

Thermal Extremes

This section does a good job at breaking down and describing the different trends and correlations between in situ and SSST measurements for maximum and minimum temperatures. However, it needs to put more into context what this means for coral reefs – while it is briefly mentioned at the very end of the last paragraph, this could use a bit more emphasis and context to highlight the relevance, considerations, and implications.

Thermal Variability

This is a very relevant section ecologically; however, it feels like it came out of nowhere. Thermal variability, its definition, and importance for coral reefs was not mentioned prior to this section – it needs to be described in the Introduction section. There are also not methods described for this section in the Methos section – they are only described here in one sentence. Also, it is stated that “diurnal temperature” was used to assess whether satellite-derived temperatures can be used to infer variability, but this feels like an “apple to oranges” comparison. They are expected to be different as they are collected at different times in the day (day vs night). I’m not saying I consider the analysis to be “wrong” (for lack of a better word), there might just be a better way to frame this to make the suitability of the comparison clear.

Heat Accumulation & Bleaching Risk

This section is very valuable and relevant for management. But, just like the section above, some terms need to be introduced earlier, the methods need to be described in a bit more detail in the Methodology section, and the discussion could be expanded upon.

Conclusion

This section does a good job at summarizing the key findings and putting them into context. Some rewording is needed, particularly for avoiding repetition.

Line Specific Comments

59-60: This sentence repeats the example already listed in line 57 where “monitoring coral bleaching risk globally” is mentioned. Consider adding heat stress to the list of examples in the previous sentence (with an appropriate reference), or separating the claim from the examples in the previous sentence.

68: The word “miss” suggests there is no data for these features/regions. I think it is more about the large difference between reef feature size and SST pixel size (a main point of the manuscript). Consider rewording this sentence to something like “Second, often the difference between reef features size and SST pixel size (>1km²) is too large to retrieve detailed, accurate information about the thermal stress a reef is experiencing”.

70-71: It would be good to provide references for this. The term “skin” might not be necessary as it is not referred to at any other point in the manuscript. Consider adding a short statement as to why “temperatures at depth” are important (e.g. “…which are more representative of the thermal stress that corals actually experience (add a reference).)”.

72: Consider changing the word “issues” for “limitations”.

79-83: This sentence needs to be reworded.

83-86: What is meant by “scales”? It would also read easier if “and are” is replaced by “and is” – referring to the metric.

91: “Datasets” is one word (inconsistent throughout the manuscript). Not sure if “deployed on local reefs” is necessary.

92-94: It would be good to add refences here.

100-104: Consider this sentence at “in particular”. Add a reference at the end.

Figure 1: Where is the bathymetry layer sources from? (add reference for this). Consider moving up the figure under the paragraph where it is first mentioned (this for all figures).

137: what is meant by “having multiple loggers per satellite”?

154-158: This sentence can be split as it is dense and hard to follow. Also, be careful of the use of the word “significance”. What I assume you mean to say is that if the intercept does not differ significantly from 0, and the slope does not differ significantly from 1, then there is no bias, and the SST and in situ measurements are consistent. The part of this sentence after “otherwise” can be a separate sentence and needs to be reworded – I think “on the same time scale” might not be necessary and makes the sentence a bit confusing.

176: In the methods you say the data is from November 2017 to January 2020. The dates of the analyses you list here are different (2018-2021). Did you discard the two months of 2017 (Nov/Dec)? Where did the rest of the data for 2020 and 2021 come from? Are there typos somewhere with the dates?

179: You list Table 1, but I can’t find it. Or is this meant to be Table S1?

Figure 2 and 3: Figure 3 is mentioned for the first time in line182, and Figure 2 in line 199. These should be rearranged to follow in the right order.

223: This part of the sentences is missing a connecting word. Maybe “… but more extreme and frequent difference between regions is displayed by in situ data”.

256-258: This sentences and the sentence in lines 253-256 could be merged into one and made more concise as the finding is the same for the regional and “all” analyses (like lines 262-263 for minimum temperatures).

299-307: These lines read like the belong more in the introduction.

308: Is there a citation for the methodology? Where can this be found? Maybe this should be included/expanded upon in the methodology section.

315-319: Same as above, maybe this should be included in the methodology section?

331-344: I see the bulk of the discussion lies in this paragraph. I think it can be expanded upon. Can you add discussion about timescales, e.g., what is described in lines 322-323 and 323-325? What does this mean for post-disturbance recovery and potential shifts in benthic communities? Maybe there could be some brief discussion about the lack of ability to determine even finer scales (within-reef, i.e., geomorphic zones) variation, where for example reef flats and reef slopes do not always respond the same way to the same thermal stress event? Look at the work done on Heron Reef, Australia, on the bleaching in 2020 for example https://doi.org/10.1007/s00338-022-02328-6.

Figure 6: The labels in the x-axes are a bit hard to see as they are clumped together. Consider putting them vertically, and maybe even putting them only every 2/3 months.

371-378: These sentences feel a bit repetitive. Consider condensing them.

382-383: Do you mean “Coral reef restoration and mitigation…”?

Reviewer #2: The study presents a valuable comparison between satellite-derived sea surface temperatures and in situ temperature data from coral reefs in Palau, highlighting important limitations of SSST in capturing physiologically relevant thermal metrics. The research is well-motivated, the methodology is sound, and the findings have significant implications for coral reef management and restoration efforts.

Below, I have summarized my primary comments and recommendations. My final decision is Accept with Major Corrections, pending satisfactory revisions.

General Comments:

1. Introduction: Well-written and effectively sets the context for the study.

2. Keywords: The current keywords are redundant (e.g., "Coral Reef", "Sea Surface Temperature", and "Palau" appear in the title) and "in situ" is too generic. I recommend more specific and non-redundant keywords.

3. Model Description (Lines 145–169): The statistical model is adequately described but could be more clearly illustrated. I suggest including a flowchart to enhance clarity and conciseness.

4. Figures: Several figures require improvement to better communicate the study’s findings. Specific recommendations are provided below.

5. More statistical analysis needed

Specific Comments on Figures:

Figure 1 (Site Map):

- The inset map does not clearly show Palau’s location in the broader Pacific region.

- Bathymetry colors and lines partially obscure the data collection sites.

- Coral reef patches or sampling locations should be more prominently highlighted.

Figure 2:

- The current scatterplot with regression coefficients does not fully convey the statistical performance of SSST vs. in situ data.

- I recommend using box plots to display statistical metrics such as RMSE, MAPE, bias, etc., for better visual comparison.

Figure 3:

- The current layout is not sufficiently informative.

- I suggest reorganizing the panels in rows with wider areas to better show temporal fluctuations.

- Adding a weekly/monthly moving average line and shading seasonal periods would improve interpretability.

Figure 4:

- Axis labels are unclear. The X-axis should be labeled more descriptively (e.g., "Temperature Difference (°C)").

- The Y-axis label and the meaning of the plots need clarification.

- Each subplot should be clearly labeled with the corresponding regional comparison.

Figure 6:

- Panels (a) and (b): Improve X-axis labels and use thinner lines for better clarity.

- Panel (c): The trend line is not clearly visible. Enhance its prominence for better interpretation.

**Do you want your identity to be public for this peer review?** For information about this choice, including consent withdrawal, please see our Privacy Policy

Reviewer #1: **Yes:** David E. Carrasco Rivera

Reviewer #2: **Yes:** Masoud Moradi

---

## [Author Response · Author response to Decision Letter 1]

2 Jan 2026

Reviewer #1: Overview

Overall, this manuscript presents an interesting and valuable story about comparing in situ and satellite-derived thermal information and their implications for coral reef management. The major claim of this paper relates to the limitations of satellite-derived thermal measurements to provide information at management-relevant scales. The manuscript is well written and flows in a way that is easy to follow and understand. However, there are a few things that need some attention. First, the title suggests that reef physiology (e.g., accretion, benthic composition, primary production, etc.) is examined in the manuscript as a response variable, but this is not really the case – it’s more about thermal stress and temperature as an environmental variable at different scales. Second, some terms that are discussed in the results/discussion section need to be presented earlier both in the introduction and methodology sections. Some rewording is also necessary across the different sections to reduce repetition and increase clarity.

I recommend the manuscript to be considered for publication once the revisions have been incorporated.

Introduction

This section does a good job at highlighting the relevance of the research study and putting it into context. Some rewording and adding references to support some claims would be beneficial.

Thank you. We reworded portions of the text and inserted more supporting references to improve the clarity and content.

Methods

This section is very detailed and provides enough information regarding data origin and management. It also provides good information about the statistical analyses. However, some rewording is needed as robust analytical methods are presented and sometimes become hard to follow.

Thank you. We reworded part of the statistics section in the methods to improve clarity and flow. We also inserted a flowchart to help clarify data sources and which models were applied to which thermal metrics.

More detail about the composition in terms of reef characteristics (reef types, geomorphology, overall benthic composition/communities, etc.) might help to understand the differences between the regions (Northern, Southern, Western) and put into perspective the importance of finer-scale thermal data.

We have inserted a few sentences detailing reef type and benthic cover of the regions and how the regions differ in exposure/water flow.

Results/Discussion

Overall, this section does a good job at presenting and summarizing the main findings of the study. However, if results and discussion are joint into one single section, the authors need to ensure the results are discussed more thoroughly and put into context as they are presented.

Thank you. We have expanded the discussion in multiple sections (detailed below).

Average temperatures

This section does a good job at summarizing the comparison between average in situ and SSST measurements. However, it needs to put more into context what these differences represent: 0.39/0.58/0.71/1°C difference seems small, but is it?

What does it mean for coral reefs and managers?

Thank you. We added text and references to elaborate on why that difference, while seemingly small, can lead to meaningful differences and consequences.

Inter-regional Comparisons

This section does a good job at highlighting the differences across regions, but, as the section before, it needs to put discuss what this means for coral reefs and for management.

Thank you. We added a discussion on how these regional differences are particularly important to coral restoration groups that rely on accurate temperature trends to select sites for their in situ coral nurseries.

Thermal Extremes

This section does a good job at breaking down and describing the different trends and correlations between in situ and SSST measurements for maximum and minimum temperatures. However, it needs to put more into context what this means for coral reefs – while it is briefly mentioned at the very end of the last paragraph, this could use a bit more emphasis and context to highlight the relevance, considerations, and implications.

Thank you. We expanded discussion in this section to include points on how missed maxima can lead to underestimation of heat accumulation.

Thermal Variability

This is a very relevant section ecologically; however, it feels like it came out of nowhere. Thermal variability, its definition, and importance for coral reefs was not mentioned prior to this section – it needs to be described in the Introduction section. There are also not methods described for this section in the Methods section – they are only described here in one sentence.

We have added description of thermal variability in introduction and methods sections, along with more detail on why this is an important metric to consider and how it is derived.

Also, it is stated that “diurnal temperature” was used to assess whether satellite-derived temperatures can be used to infer variability, but this feels like an “apple to oranges” comparison. They are expected to be different as they are collected at different times in the day (day vs night). I’m not saying I consider the analysis to be “wrong” (for lack of a better word), there might just be a better way to frame this to make the suitability of the comparison clear.

We have added some more detail to help clarify why, even if the comparison may seem inappropriate, it is important to see if SSST can be used in any capacity to gauge thermal variability.

Heat Accumulation & Bleaching Risk

This section is very valuable and relevant for management. But, just like the section above, some terms need to be introduced earlier, the methods need to be described in a bit more detail in the Methodology section, and the discussion could be expanded upon.

We added more detail on how exactly DHW is calculated in the methods section. We also greatly expanded the discussion to address how heat accumulation affects corals and the down-stream effects of the consequences of misidentification of bleaching risk/heat accumulation.

Conclusion

This section does a good job at summarizing the key findings and putting them into context. Some rewording is needed, particularly for avoiding repetition.

Thank you. We rephrased portion of the beginning and end of the conclusion to improve clarity and remove repetitive phrases.

Line Specific Comments

59-60: This sentence repeats the example already listed in line 57 where “monitoring coral bleaching risk globally” is mentioned. Consider adding heat stress to the list of examples in the previous sentence (with an appropriate reference), or separating the claim from the examples in the previous sentence.

Heat stress added to previous sentence with reference.

68: The word “miss” suggests there is no data for these features/regions. I think it is more about the large difference between reef feature size and SST pixel size (a main point of the manuscript). Consider rewording this sentence to something like “Second, often the difference between reef features size and SST pixel size (>1km²) is too large to retrieve detailed, accurate information about the thermal stress a reef is experiencing”.

This sentence has been reworded.

70-71: It would be good to provide references for this. The term “skin” might not be necessary as it is not referred to at any other point in the manuscript. Consider adding a short statement as to why “temperatures at depth” are important (e.g. “… which are more representative of the thermal stress that corals actually experience (add a reference).)”.

Removed the term “skin” and added explanatory clause and reference.

72: Consider changing the word “issues” for “limitations”.

Corrected.

79-83: This sentence needs to be reworded.

Reworded – split into 2 sentences.

83-86: What is meant by “scales”? It would also read easier if “and are” is replaced by “and is” – referring to the metric.

Addressed and reworded as suggested.

91: “Datasets” is one word (inconsistent throughout the manuscript). Not sure if “deployed on local reefs” is necessary.

Corrected

92-94: It would be good to add refences here.

Reference added

100-104: Consider this sentence at “in particular”. Add a reference at the end.

Split into two sentences and added references on important thermal metrics that help determine coral resilience.

Figure 1: Where is the bathymetry layer sources from? (add reference for this). Consider moving up the figure under the paragraph where it is first mentioned (this for all figures).

Added reference for bathymetry layers and adjusted figure locations.

137: what is meant by “having multiple loggers per satellite”?

We reworded the sentence to clarify that in each SSST block/grid cell we have multiple in situ loggers.

154-158: This sentence can be split as it is dense and hard to follow. Also, be careful of the use of the word “significance”. What I assume you mean to say is that if the intercept does not differ significantly from 0, and the slope does not differ significantly from 1, then there is no bias, and the SST and in situ measurements are consistent. The part of this sentence after “otherwise” can be a separate sentence and needs to be reworded – I think “on the same time scale” might not be necessary and makes the sentence a bit confusing.

We split up the sentences and replaced the phrase “on the same time scale” to improve clarity.

176: In the methods you say the data is from November 2017 to January 2020. The dates of the analyses you list here are different (2018-2021). Did you discard the two months of 2017 (Nov/Dec)? Where did the rest of the data for 2020 and 2021 come from? Are there typos somewhere with the dates?

Corrected typos in dates

179: You list Table 1, but I can’t find it. Or is this meant to be Table S1?

Yes, corrected

Figure 2 and 3: Figure 3 is mentioned for the first time in line182, and Figure 2 in line 199. These should be rearranged to follow in the right order.

Corrected

223: This part of the sentences is missing a connecting word. Maybe “… but more extreme and frequent difference between regions is displayed by in situ data”.

Corrected

256-258: This sentences and the sentence in lines 253-256 could be merged into one and made more concise as the finding is the same for the regional and “all” analyses (like lines 262-263 for minimum temperatures).

Rephrased sentences to make it more brief and less repetitive.

299-307: These lines read like the belong more in the introduction.

Rearranged and incorporated into introduction

308: Is there a citation for the methodology? Where can this be found? Maybe this should be included/expanded upon in the methodology section.

Citation and content added to description of DHW in methods section

315-319: Same as above, maybe this should be included in the methodology section?

Added explanatory section and citation in methods

331-344: I see the bulk of the discussion lies in this paragraph. I think it can be expanded upon. Can you add discussion about timescales, e.g., what is described in lines 322-323 and 323-325? What does this mean for post-disturbance recovery and potential shifts in benthic communities? Maybe there could be some brief discussion about the lack of ability to determine even finer scales (within-reef, i.e., geomorphic zones) variation, where for example reef flats and reef slopes do not always respond the same way to the same thermal stress event? Look at the work done on Heron Reef, Australia, on the bleaching in 2020 for example https://doi.org/10.1007/s00338-022-02328-6.

We have added more discussion about different reef types and thermal variability, and incorporated the work done on Heron Reef.

Figure 6: The labels in the x-axes are a bit hard to see as they are clumped together. Consider putting them vertically, and maybe even putting them only every 2/3 months.

Adjusted axis labels and corrected date spacing.

371-378: These sentences feel a bit repetitive. Consider condensing them.

We have removed redundant text in these sentences for brevity

382-383: Do you mean “Coral reef restoration and mitigation…”?

We intended this comment to apply to restoration efforts in other marine ecosystems as well, but have added “marine” to clarify the scope.

Reviewer #2:

The study presents a valuable comparison between satellite-derived sea surface temperatures and in situ temperature data from coral reefs in Palau, highlighting important limitations of SSST in capturing physiologically relevant thermal metrics. The research is well-motivated, the methodology is sound, and the findings have significant implications for coral reef management and restoration efforts. Below, I have summarized my primary comments and recommendations. My final decision is Accept with Major Corrections, pending satisfactory revisions.

General Comments:

1. Introduction: Well-written and effectively sets the context for the study.

Thank you.

2. Keywords: The current keywords are redundant (e.g., "Coral Reef", "Sea Surface Temperature", and "Palau" appear in the title) and "in situ" is too generic. I recommend more specific and non-redundant keywords.

Altered key words to be more specific. Removed redundant terms.

3. Model Description (Lines 145–169): The statistical model is adequately described but could be more clearly illustrated. I suggest including a flowchart to enhance clarity and conciseness.

We have added a flowchart (now figure 2) detailing data sources, derived quantities, and statistical analyses.

4. Figures: Several figures require improvement to better communicate the study’s findings. Specific recommendations are provided below.

Thank you – we have multiple figures to improve clarity and visual flow as detailed in our responses below.

5. More statistical analysis needed

To further examine variation between data origins (in situ and SSST) and regions, we have added two multivariate state-space models. These models allowed for a stronger comparison of the data types: for instance, they show that much of the temporal variation in nightly average temperature can be explained by shared trends within data origins, i.e., that variation in in situ data in each region more closely mirrors in situ data in other regions than collocal SSST measurements. They also allow for explicit partitioning of variance in each time series into different sources. Additionally, we have added a plot displaying multiple fit and prediction statistics (R2, RMSE, MAE, and bias) per your figure suggestions.

Specific Comments on Figures:

Figure 1 (Site Map):

- The inset map does not clearly show Palau’s location in the broader Pacific region.

- Bathymetry colors and lines partially obscure the data collection sites.

- Coral reef patches or sampling locations should be more prominently highlighted.

We have made multiple changes to this figure to improve legibility: (1) the inset map is now larger and centered on Palau’s location in the Indo-Pacific, (2) bathymetry line colors have been changed to improve contrast with site points, (3) point and regional bounding box colors have been darkened, white borders have been added around regional bounding boxes, and the width of existing white borders around site points have been increased, and (4) the map has been zoomed to increase the size of all elements.

Figure 2: data.

- The current scatterplot with regression coefficients does not fully convey the statistical performance of SSST vs. in situ

- I recommend using box plots to display statistical metrics such as RMSE, MAPE, bias, etc., for better visual comparison.

We have added a figure with fit statistics for each of the regressions of in situ temperature data on SSST. Because there are only four regressions (one for each region) for each metric (nightly mean, nightly max, nightly min, and monthly mean), and we felt that it was inappropriate to combine fit statistics for different metrics into a single box plot, we opted instead to plot individual points for fit statistics of each region by

---

## [Editor Report · Decision Letter 1]

14 Jan 2026

Satellite-derived temperature measures miss key physiologically relevant thermal trends on Palauan reefs

PONE-D-25-48462R1

Dear Dr. Lippert,

We’re pleased to inform you that your manuscript has been judged scientifically suitable for publication and will be formally accepted for publication once it meets all outstanding technical requirements.

Kind regards,

Parviz Tavakoli-Kolour

Academic Editor

PLOS One

---

## [Editor Report · Acceptance letter]

PONE-D-25-48462R1

PLOS One

Dear Dr. Lippert,

I'm pleased to inform you that your manuscript has been deemed suitable for publication in PLOS One. Congratulations! Your manuscript is now being handed over to our production team.

Kind regards,

on behalf of

Dr. Parviz Tavakoli-Kolour

Academic Editor

PLOS One